# Implementation of Feed Efficiency in Iranian Holstein Breeding Program

**DOI:** 10.3390/ani13071216

**Published:** 2023-03-31

**Authors:** Sara Nadri, Ali Sadeghi-Sefidmazgi, Pouya Zamani, Gholam Reza Ghorbani, Sajjad Toghiani

**Affiliations:** 1Department of Animal Sciences, College of Agriculture, Isfahan University of Technology, Isfahan 83111-84156, Iran; sara.nadri22@gmail.com (S.N.);; 2Department of Animal Science, University of Tehran, Karaj P.O. Box 3158711167-4111, Iran; 3Department of Animal Science, Faculty of Agriculture, Bu-Ali Sina University, Hamedan 65176-58978, Iran; 4Animal Genomics and Improvement Laboratory, Agricultural Research Service, USDA, Beltsville, MD 20705-2350, USA; sajjad.toghiani@usda.gov

**Keywords:** feed efficiency, meta-analysis, economic weight, selection index

## Abstract

**Simple Summary:**

Including feed efficiency in breeding objectives for dairy cattle is desirable due to the potential benefit for increased sustainability and herd profitability. An investigation of selection indices that included the direct and indirect selection of feed efficiency traits showed that the best total response was observed when residual feed intake was directly included as a selection criterion. Using indicator traits (e.g., type) could be a useful proxy in the absence of direct genetic evaluations for feed efficiency. Breeding objectives of the current Iranian selection index, which currently focuses on future production and marketing environment, could include feed efficiency in future evaluations.

**Abstract:**

This study aimed to evaluate the economic impact of improving feed efficiency on breeding objectives for Iranian Holsteins. Production and economic data from seven dairy herds were used to estimate the economic values of different traits, and a meta-analysis was conducted to analyze the genetic relationships between feed efficiency and other traits. Economic weights were calculated for various traits, with mean values per cow and per year across herds estimated at USD 0.34/kg for milk yield, USD 6.93/kg for fat yield, USD 5.53/kg for protein yield, USD −1.68/kg for dry matter intake, USD −1.70/kg for residual feed intake, USD 0.47/month for productive life, and USD −2.71/day for days open. The Iranian selection index was revised to improve feed efficiency, and the feed efficiency sub-index (FE$) introduced by the Holstein Association of the United States of America was adopted to reflect Iran’s economic and production systems. However, there were discrepancies between Iranian and US genetic coefficients in the sub-index, which could be attributed to differences in genetic and phenotypic parameters, as well as the economic value of each trait. More accurate estimates of economic values for each trait in FE$ could be obtained by collecting dry matter intake from Iranian herds and conducting genetic evaluations for residual feed intake.

## 1. Introduction

Feed costs account for 40 to 60% of the total costs associated with milk production [1]. For the dairy industry, collecting sufficient information on feed intake to enable the development and implementation of genetic evaluations for feed efficiency was and remains an expensive and difficult task [2]. However, efforts to include feed efficiency as part of dairy breeding objectives began in the 1990s because feed efficiency has a significant impact on the profitability of dairy production. Feed efficiency traits have been studied from various perspectives in the USA, Australia, Canada, Netherlands, Denmark, Sweden, and Finland, including estimation of economic value [3], breeding value [2], genetic evaluation [4,5], and finally inclusion in breeding objectives [6]. Improved feed efficiency is also desirable due to its potential benefits toward a sustainable environment by reducing greenhouse gas emissions per animal as well as farm profitability by reducing the input of natural resources for equivalent output production [7,8].

Different traits have been used to study genetic diversity for feed efficiency in dairy cattle. Feed intake measured as DMI is a key component for calculating feed efficiency in dairy cattle, and its heritability has been reported as being in a range from 0.08 to 0.52 [9,10,11]. Residual feed intake (RFI) is a valuable metric and preferred trait for feed efficiency; it is estimated from DMI and is defined as the difference between observed and predicted feed intake after accounting for body size, body weight change, and milk production [12]. With the concept of RFI [13], the definitions of residual methane (adjusting methane for production, feed, and weight) have been suggested [14,15,16] to explain express emission in terms of CH4 per kg of output, as CH4 intensity (MeI = g CH4/kg milk), or input such as methane yield (MeY = g CH4/kg feed). As RFI has been genetically adjusted for the main energy sinks (e.g., milk yield, BW), it is genetically uncorrelated with those energy sink traits. Milk production and BW-related traits are often part of the total merit index traits in the genetic evaluation of dairy cattle. Thus, after eliminating the genetic correlations of RFI with milk production and BW, RFI becomes a more independent trait for representing FE in the selection index [17]. The estimated heritability of RFI ranges from 0.14 to 0.38 in dairy cows [18,19,20]. The heritability of RFI depends largely on the environmental correlation between feed intake and milk production. RFI is dependent and independent of production, based on the phenotypic and genotypic regression of feed intake on milk production, respectively [21].

Breeding objectives and selection indices define the direction of selection in a livestock population. One of the most important tasks in developing an economic selection index is to estimate the economic values (EVs) of specific traits because these EVs depend on variety of revenue and expense variables [22]. Therefore, selection index coefficients for different traits must be quantified to assess an economic selection index. Calculation of RFI EVs in dairy cattle is complicated due to the dynamic changes in body weight (BW) and body condition score (BCS) during the lactation cycle [23]. The commonly used RFI model employs constant partial regressions of feed intake on energy sinks. Considering the complexity of the metabolic changes in cows across lactation stages, it is possible that the partial regression coefficients of feed intake on milk production, body maintenance, and BW change may differ across lactation stages. Therefore, the general RFI model with constant partial regression coefficients of feed intake on energy sinks may not always hold for the entire lactation period, influencing RFI estimation [17]. In addition, the lack of a cost-efficient method for measuring RFI on commercial dairy farms has limited its inclusion in breeding goals for dairy production systems. The main obstacle to integrating feed efficiency traits in breeding programs for dairy cattle is the restricted phenotypic database, which results in low genomic reliability. Thus, optimizing the collection of feed intake data from multiple sources is needed to increase the reference population size so that genomic prediction accuracy for feed efficiency can be improved [20].

The direct measurement of DMI and RFI currently is not possible in the Iranian dairy production system. Therefore, using economically important traits such as type traits with a moderate-to-high correlation with DMI and RFI is an alternative option. In addition, to improve genetic gain, Iran imports dairy bull semen from countries that include feed efficiency traits in their selection indices such as the USA, Canada, Germany, and the Netherlands.

Although feed efficiency traits are not documented in Iran, the genetic difference between cows for feed intake provides an opportunity to reduce costs through genetic selection. However, the estimation of economic coefficients for various traits of Iranian dairy cows is needed to form an economic selection index. The objectives of this study were (1) to investigate the genetic analysis of feed efficiency traits (DMI and RFI) of Holstein cows using meta-analysis, (2) to develop a bioeconomic model to estimate the EVs of DMI and RFI, and (3) to revise the current Iranian selection index for Holstein dairy cows to improve feed efficiency.

## 2. Materials and Methods

### 2.1. Meta-Analysis

Quantitative findings of numerous studies have been collected and integrated using meta-analysis techniques to learn more about the statistical significance and relevance of feed efficiency traits. This quantitative technique uses specific measures to indicate the strength of variable relationships for studies included in the analysis [24]. The genetic parameters used in this study (122 heritabilities and 95 genetic correlations) originated from a comprehensive review of 37 scientific studies that were published between 1991 and 2021. Those studies estimated heritability and genetic and phenotypic correlations for feed intake (DMI and RFI), production (milk, fat, and protein yields; MY, FY, and PY, respectively), reproduction (days open; DO), longevity (productive life; PL), type (stature, chest width, body depth, angularity, rump angle, and rump width), BW, and BCS traits of Holstein dairy cows in different countries.

*Pooled Heritability*. Koots’ method was used to calculate the weighted mean of heritability [25], and pooled heritability ($h_{\mathrm{pooled}}^{2}$) was
$$h_{\mathrm{pooled}}^{2}=\frac{\sum_{i=1}^{n}(h_{i}^{2}/{\mathrm{SE}}_{h_{i}^{2}}^{2})}{\sum_{i=1}^{n}\left(1/{\mathrm{SE}}_{h_{i}^{2}}^{2}\right)}$$
where n is the number of published studies, $h_{i}^{2}$ is the heritability estimate from published study i, and SE is standard error calculated as the weighted mean of standard error divided by the square root of the number of records; weighted mean of standard error was calculated as follows [26]:$$\mathrm{SE}=\sqrt{\frac{\sum_{i=1}^{n}\sigma_{i}^{2}x_{i}^{2}}{\sum_{i=1}^{n}x_{i}}}$$
where $\sigma_{i}^{2}$ is sampling variance and $x_{i}$ is a number of records. 

*Pooled Correlations.* Meta-analysis of genetic correlations among traits of reported values in the literature was determined by transforming those values to an approximate normal scale using Fisher’s *Z* transformation [27]:$$Z=0.5\;log\left(\frac{1+r_{ij}}{1-r_{ij}}\right)$$
$${\mathrm{SE}}_{Z}=\sqrt{1/\left(n-3\right)}$$
$$Z^{\prime }=\overset{x}{\underset{i=1}{\sum }}\left(Z_{i}/{\mathrm{SE}}_{Z_{i}}^{2}\right)$$
$$Z_{\mathrm{pooled}}=\frac{Z^{\prime }}{\sum_{i=1}^{x}\left(1/{\mathrm{SE}}_{Z_{i}}^{2}\right)}$$
and
$$r_{g}=\left(e^{2Z}-1\right)/\left(e^{2Z}+1\right)$$
where $r_{ij}$ is the genetic correlation between traits *i* and *j*; *x* is the number of records observed per trait; and $Z^{\prime }$ is the weighted mean for the *Z*-transformed genetic correlations, which is used to estimate the pooled genetic correlation ($Z_{\mathrm{pooled}}$) and back-transformed to obtain the weighted mean genetic correlation ($r_{g}$). This methodology assumes a genetic correlation of 1 between traits in similar target groups, such as production, reproduction, longevity, and feed intake.

### 2.2. Economic Weights

This study focused on seven Holstein dairy herds that were selected as typical examples of Iranian herds with respect to milk production, reproduction, culling, and productive life cows. These farms implemented a management, feeding, and housing system that is commonly used across the majority of Iranian dairy herds. All of the participating farms were systematically recorded by the Animal Breeding Center of Iran and were located in four Iranian provinces (Isfahan, Tehran, Qazvin, and Razavi Khorasan), which are known for housing the majority of industrial dairy farms. Table 4 provides descriptive statistics on production data and variables utilized to compute EVs. Market conditions in 2020 were the basis for generating the input parameters to derive EV. All costs and prices were converted from Iranian rial currency (IRR) to US currency (USD) using a conversion rate of 1 USD = 42,000 IRR.

The EVs were estimated based on the Iranian data from the 7 large Holstein dairy herds for each trait group (production, longevity, reproduction, and feed intake) using the bioeconomic model described in [28]. The economic weights (EWs) were calculated by multiplying the EV of each trait by cumulative discounted expressions (CDEs). The term CDEs refers to a correction for the accumulation of values over time, where “cumulative” refers to this accumulation. The term “discounted” is used to describe the process of adjusting future returns to reflect their present-day value [29].

The economic value of DMI was expressed in kg of dry matter intake in each lactation based on production and body size. The DMI was estimated as a function of DIM, BW, and corrected 3.2% milk fat [30]. The cost of 1 kg of DMI was calculated and compared using 4 different scenarios based on parity, production level, forage-to-concentrate (F:C) ratio, and concentrate source. Diets were formulated using Cornell Net Carbohydrate and Protein System software (CNCPS, version 5.0; Cornell University, Ithaca, NY, USA) [31]. The diet for the first parity was designed with a single F:C ratio of 45:55. For the second parity and later, the F:C ratio was 40:60, and the F:C ratio for high- (above 40 kg), moderate- (around 35 kg), and low-production (below 25 kg) cows was 35:65, 48:52, and 50:50, respectively. Two different concentrate sources were investigated for each of the cow groupings. 

The definition of RFI and predicted daily DMI for dairy cows was determined using nutrient requirements of the Iranian production system with respect to net energy and metabolizable protein based on the equations found in [30]. The required amounts of net energy, metabolizable protein, and DMI were 4.55 (Mcal/kg of DMI), 121 (Mcal/kg of DMI), and 22.6 (kg) for a high-production cow with a BW of 680 kg and an average milk yield of 40 kg/day, respectively. Mean RFI was calculated from the difference between actual and predicted daily DMI. The EV for feed efficiency traits was defined as the partial derivative of the bioeconomic model with respect to the DMI and RFI traits included in the breeding objective.

The following relationship was used to calculate the economic value.
$$\mathrm{EV}=\frac{TP_{\mathrm{high}}-TP_{low}}{TV_{\mathrm{high}}-TV_{low}}$$
where $TV_{high}$ and $TV_{low}$ represent increases and decreases in the value of trait *i*, respectively; $TP_{\mathrm{high}}$ and $TP_{low}$ represent the total profit in the evaluated production system that was considered if the average attribute *i* increased or decreased, respectively.

### 2.3. Breeding Objectives

Traits of economic importance for MY, FY, PY, PL, DO, DMI, and RFI were defined in the Iranian dairy breeding program to explore 2 different breeding objectives ($H_{1}$ includes DMI while $H_{2}$ includes RFI). Both $H_{1}$ and $H_{2}$ breeding objectives were considered to investigate the impact of including feed efficiency traits (DMI and RFI) on the genetic and economic gains. 

Multiplying the EW of each trait by the EBV of its corresponding trait gives the aggregate breeding value (breeding objective):$$H_{1}={\mathrm{EW}}_{\mathrm{MY}}\left({\mathrm{EBV}}_{\mathrm{MY}}\right)+{\mathrm{EW}}_{\mathrm{FY}}\left({\mathrm{EBV}}_{\mathrm{FY}}\right)+{\mathrm{EW}}_{\mathrm{PY}}\left({\mathrm{EBV}}_{\mathrm{PY}}\right)+\;\;\;{\mathrm{EW}}_{\mathrm{PL}}\left({\mathrm{EBV}}_{\mathrm{PL}}\right)+{\mathrm{EW}}_{\mathrm{DO}}\left({\mathrm{EBV}}_{\mathrm{DO}}\right)+{\mathrm{EW}}_{\mathrm{DMI}}\left({\mathrm{EBV}}_{\mathrm{DMI}}\right)$$
$$H_{2}={\mathrm{EW}}_{\mathrm{MY}}\left({\mathrm{EBV}}_{\mathrm{MY}}\right)+{\mathrm{EW}}_{\mathrm{FY}}\left({\mathrm{EBV}}_{\mathrm{FY}}\right)+{\mathrm{EW}}_{\mathrm{PY}}\left({\mathrm{EBV}}_{\mathrm{PY}}\right)+\;\;\;{\mathrm{EW}}_{\mathrm{PL}}\left({\mathrm{EBV}}_{\mathrm{PL}}\right)+{\mathrm{EW}}_{\mathrm{DO}}\left({\mathrm{EBV}}_{\mathrm{DO}}\right)+{\mathrm{EW}}_{\mathrm{RFI}}\left({\mathrm{EBV}}_{\mathrm{RFI}}\right)$$

### 2.4. Selection Index Scenarios

Several indices were established to examine the genetic and economic responses to selection. Effect of breeding objective on response per trait per generation was investigated using the selection index theory [32]. The optimal selection index coefficients (***b***) for a set of phenotypic traits maximize the correlation of phenotypic index ***I*** with breeding objectives H. To establish distinct selection criteria for each index scenario, the phenotypic values of MY, FY, PL, DO, DMI, and RFI for bulls with at least 60 daughters with a record for each trait were examined. The calculation of selection index with several traits required variability, heritability, relative EVs, and correlations among their phenotypes and genotypes.

A total of 9 different selection indices (***I***) have been chosen to investigate the direct and indirect response to selection based on heritabilities and their genetic correlations used from the meta-analysis for study feed efficiency traits. For the first scenario, 7 different selection indices based on the direct selection of DMI and RFI were developed and compared for simplicity and effectiveness in determining genetic and economic trends:$$\begin{array}{c} I_{1}=b_{\mathrm{MY}}\left(\mathrm{MY}\right)+b_{\mathrm{FY}}\left(\mathrm{FY}\right)+b_{\mathrm{PY}}\left(\mathrm{PY}\right)+b_{\mathrm{PL}}\left(\mathrm{PL}\right)+b_{\mathrm{DO}}\left(\mathrm{DO}\right)\\ I_{2}=b_{\mathrm{MY}}\left(\mathrm{MY}\right)+b_{\mathrm{FY}}\left(\mathrm{FY}\right)+b_{\mathrm{PY}}\left(\mathrm{PY}\right)+b_{\mathrm{PL}}\left(\mathrm{PL}\right)+b_{\mathrm{DO}}\left(\mathrm{DO}\right)+b_{\mathrm{DMI}}\left(\mathrm{DMI}\right)\\ I_{3}=b_{\mathrm{MY}}\left(\mathrm{MY}\right)+b_{\mathrm{FY}}\left(\mathrm{FY}\right)+b_{\mathrm{PY}}\left(\mathrm{PY}\right)+b_{\mathrm{PL}}\left(\mathrm{PL}\right)+b_{\mathrm{DO}}\left(\mathrm{DO}\right)+b_{\mathrm{RFI}}\left(\mathrm{RFI}\right)\\ I_{4}=b_{\mathrm{MY}}\left(\mathrm{MY}\right)+b_{\mathrm{FY}}\left(\mathrm{FY}\right)+b_{\mathrm{PY}}\left(\mathrm{PY}\right)+b_{\mathrm{PL}}\left(\mathrm{PL}\right)+b_{\mathrm{BW}}\left(\mathrm{BW}\right)+b_{\mathrm{DMI}}\left(\mathrm{DMI}\right)\\ I_{5}=b_{\mathrm{MY}}\left(\mathrm{MY}\right)+b_{\mathrm{FY}}\left(\mathrm{FY}\right)+b_{\mathrm{PY}}\left(\mathrm{PY}\right)+b_{\mathrm{PL}}\left(\mathrm{PL}\right)+b_{\mathrm{BCS}}\left(\mathrm{BCS}\right)+b_{\mathrm{DMI}}\left(\mathrm{DMI}\right)\\ I_{6}=b_{\mathrm{MY}}\left(\mathrm{MY}\right)+b_{\mathrm{FY}}\left(\mathrm{FY}\right)+b_{\mathrm{PY}}\left(\mathrm{PY}\right)+b_{\mathrm{PL}}\left(\mathrm{PL}\right)+b_{\mathrm{BW}}\left(\mathrm{BW}\right)+b_{\mathrm{RFI}}\left(\mathrm{RFI}\right)\\ I_{7}=b_{\mathrm{MY}}\left(\mathrm{MY}\right)+b_{\mathrm{FY}}\left(\mathrm{FY}\right)+b_{\mathrm{PY}}\left(\mathrm{PY}\right)+b_{\mathrm{PL}}\left(\mathrm{PL}\right)+b_{\mathrm{BCS}}\left(\mathrm{BCS}\right)+b_{\mathrm{RFI}}\left(\mathrm{RFI}\right) \end{array}$$
The ***b*** were calculated as $b=P^{-1}Ga$, where ***P*** is a *t* × *t* phenotypic (co)variance matrix of correlated traits, **G** is a *t* × *m* genetic (co)variance matrix for *m* traits in the breeding objective with *t* correlated traits in the selection index incorporating the additive genetic relationships between sources of information, and **a** is an *m ×* 1 vector of relative EWs in the breeding objective. The predicted response to selection for trait *m* in the breeding objective was calculated based on the method outlined by [33]
$$s\left(b^{\prime }G\right)/\sqrt{b^{\prime }Ga}$$
where ***s*** is selection intensity, which is assumed to be 1.4 and equivalent to the top 20% of sires selected, and $b^{\prime }$ is a transposed vector of selection index weights. Accuracy of a selection index was also calculated as follows [33]:$$\sqrt{b^{\prime }Ga/a^{\prime }Ca}$$
where ***a*** is a vector of EW for *m* traits in a breeding objective and ***C*** is a *m × m* genetic covariance matrix among the traits in the breeding objective.

For the second scenario, indirect or indicator traits were included as selection criteria to improve feed efficiency. In commercial dairy herds, quantification of direct feed efficiency traits is challenging. Therefore, type traits that are more heritable and simpler to measure than DMI and RFI were investigated using meta-analysis to determine the genetic correlation between the existing type traits and feed efficiency traits. The 2 type traits with the highest genetic correlations with feed efficiency traits were chest width (CW) and angularity (ANG) with DMI and CW and stature (ST) with RFI. Those correlated type traits were included as indirect indicators of DMI and RFI in 2 additional selection indices:$$\begin{array}{c} I_{8}=b_{\mathrm{MY}}\left(\mathrm{MY}\right)+b_{\mathrm{FY}}\left(\mathrm{FY}\right)+b_{\mathrm{PY}}\left(\mathrm{PY}\right)+b_{\mathrm{PL}}\left(\mathrm{PL}\right)+b_{\mathrm{DO}}\left(\mathrm{DO}\right)+b_{\mathrm{CW}}\left(\mathrm{CW}\right)+b_{\mathrm{ANG}}\left(\mathrm{ANG}\right)\\ I_{9}=b_{\mathrm{MY}}\left(\mathrm{MY}\right)+b_{\mathrm{FY}}\left(\mathrm{FY}\right)+b_{\mathrm{PY}}\left(\mathrm{PY}\right)+b_{\mathrm{PL}}\left(\mathrm{PL}\right)+b_{\mathrm{DO}}\left(\mathrm{DO}\right)+b_{\mathrm{CW}}\left(\mathrm{CW}\right)+b_{\mathrm{ST}}\left(\mathrm{ST}\right) \end{array}$$

### 2.5. Feed Efficiency Sub-Index 

The methodology and equations applied by [34] for its feed efficiency sub-index (FE$) were adapted for Iranian economics and production systems. The FE$ index is defined as the net profit that a producer receives from increased production relative to PTA for MY, FY, PY, and feed saved (FSAV). The PTA for FSAV represents the expected weight of feed saved per lactation based on BW composite and RFI evaluations. For this study, FE$ was calculated as the sum of PTA for MY, FY, PY, and FSAV weighted for EV (w):$$\mathrm{FE}\$=w_{\mathrm{MY}}\left({\mathrm{PTA}}_{\mathrm{MY}}\right)+w_{\mathrm{FY}}\left({\mathrm{PTA}}_{\mathrm{FY}}\right)+w_{\mathrm{PY}}\left({\mathrm{PTA}}_{\mathrm{PY}}\right)+w_{\mathrm{FSAV}}\left({\mathrm{PTA}}_{\mathrm{FSAV}}\right)$$
The $w_{i}$ for traits included in the Iranian FE$ formula were estimated as $b_{i,\mathrm{FE}\$}\left({\mathrm{EW}}_{i}\right)$, where $b_{i,\mathrm{FE}\$}$ is the genetic regression of FE$ on trait *i* and ${\mathrm{EW}}_{i}$ is the EW for trait *i* under the Iranian production system. The $b_{i,\mathrm{FE}\$}$ shows the change in the correlated trait as FE$ increases by 1 unit genetically and is calculated as follows:$$\;b_{i,\mathrm{FE}\$}=\frac{r_{i,\mathrm{FE}\$}\sqrt{h_{i}^{2}\sigma_{P_{i}}^{2}}}{\sqrt{h_{\;\mathrm{FE}\$}^{2}\sigma_{P_{\mathrm{FE}\$}}^{2}}}$$
where $r_{i,\mathrm{FE}\$}$ is the genetic correlation between trait i and FE$; $h_{i}^{2}$ and $h_{\mathrm{FE}\$}^{2}$ are heritabilities of trait i and FE$, respectively; and $\sigma_{P_{i}}^{2}$ and $\sigma_{P_{\mathrm{FE}\$}}^{2}$ are phenotypic variance of trait i and FE$, respectively.

The US genetic evaluations for FE$, MY, FY, PY, and FSAV released in August 2021 for 10,852 Holstein AI bulls were obtained from the Alta Genetics (Watertown, WI) website (https://bullsearch.altagenetics.com/us/BS/Advanced/HO) to calculate Iranian FE$. To obtain Iran’s FE$ at the scale of the United States, the FE$ formula for both countries (USA and Iran) was multiplied by the genetic value of that trait $\left({\mathrm{PTA}}_{i}\right)$ in proportion to the estimated coefficients of the countries themselves. Finally, a regression was used to convert the PTA for FE$ of US bulls $\left({\mathrm{PTA}}_{{\mathrm{FE}\$}_{\mathrm{USA}}}\right)$ to PTA for FE$ estimated under the Iranian production system $\left({\mathrm{PTA}}_{{\mathrm{FE}\$}_{Iran}}\right)$:(1)$${\mathrm{PTA}}_{{\mathrm{FE}\$}_{\mathrm{Iran}}}=\beta_{0}+\beta_{1}\left({\mathrm{PTA}}_{{\mathrm{FE}\$}_{\mathrm{USA}}}\right)$$
where $\beta_{0}$ is the intercept and $\beta_{1}$ is the regression coefficient.

## 3. Results

### 3.1. Meta-Analysis

Table 1 shows the weighted mean heritabilities and summary statistics for feed efficiency, production, reproduction, longevity, type, BW, and BCS traits calculated by meta-analysis across dairy cattle breeds and countries from the published literature. For feed efficiency, the weighted mean heritability estimate for DMI (0.21) was higher than that for RFI (0.19). Weighted mean heritabilities estimated for production traits were moderate (0.22) and higher than for DO (0.03) and PL (0.10). Weighted mean heritabilities for type traits ranged from 0.23 for angularity to 0.46 for stature. The weighted mean heritabilities for body size traits were estimated to be 0.23 for BCS, and 0.58 for BW. The standard errors for weighted mean heritabilities of all traits were relatively low (<0.1).

Table 2 shows the weighted mean genetic correlations and summary statistics between feed efficiency traits (DMI and RFI) and production, reproduction, longevity, type, BW, and BCS traits calculated by meta-analysis for Holsteins across countries from the published literature. Estimated weighted mean genetic correlations between DMI and production traits were strongly positive and ranged from 0.51 (FY) to 0.68 (MY). A strong positive weighted mean genetic correlation (0.49) was found between DMI and PL, whereas a weakly negative weighted mean genetic correlation (−0.14) was found between DMI and DO. Estimated weighted mean genetic correlations of DMI with type traits were all positive (0.12 to 0.58), with the largest correlations for angularity (0.58) and chest width (0.55). For RFI, weighted mean genetic correlations with production traits were weak and ranged from −0.04 for PY to 0.08 for MY. Estimated weighted mean genetic correlation for RFI was strongly negative with DO (−0.50) and moderately negative with PL (−0.23). Weighted mean genetic correlations between RFI and type traits ranged from weakly negative (−0.06) to moderately positive (0.28), with the highest correlations for stature (0.28) and chest width (0.17).

Table 3 shows the heritabilities and phenotypic and genetic correlations of feed efficiency, production, reproduction, longevity, type, BW, and BCS traits used from meta-analysis to calculate optimal weights for selection criteria in each index.

### 3.2. Economic Weights

Mean values of production data, prices, and costs of each unit for computing EVs are in Table 4. EVs, genetic expressions (genetic improvement of genes in various traits at different times on different numbers of animals), and estimated EWs for each breeding objective trait are in Table 5. An increase in MY, FY, and PY by 1 kg/cow/year results in an average EWs of USD 0.34, USD 6.93, and USD 5.53, respectively (Table 5). Mean EWs for DO and PL were USD −2.71 day/cow/year and USD 0.47 month/cow/year, respectively. The EW for 1 kg DMI/cow/year varied between herds and ranged from USD −1.43 to USD −1.88 with a mean EW of USD −1.68/cow/year. For a reduction of 1 kg DMI/day during lactation, the mean EW for RFI was estimated to be USD −1.70/cow/year.

### 3.3. Selection Index Scenarios

The expected genetic and economic gains for each breeding objective trait using different selection indices are presented in Table 6. For the first scenario when a feed efficiency trait (DMI or RFI) was included in the selection index (***I*_2_** to ***I*_7_**), genetic and economic gains were generally higher for FY and DO compared with the baseline index (*I*_1_), whereas both gains were generally lower for PL. The total selection index response when DMI or RFI was included was equal to or higher than for ***I*_1_** except for ***I*_2_**. Including DMI or RFI as a feed efficiency trait in ***I*_2_** to ***I*_7_** increased index accuracy by 1 to 8 percentage points compared to the baseline accuracy of ***I*_1_**.

For the second scenario in which type traits that were genetically correlated with feed efficiency were included in the selection index rather than DMI or RFI (***I*_8_** and ***I*_9_**), genetic and economic gains for MY, FY, PY, PL, and DO (Table 6) followed the same pattern as found for the first scenario when compared with ***I*_1_**. For DMI, higher genetic and economic gains for MY, FY, PL, and DO; lower gains for PY; higher index accuracy (2 percentage points); and higher total selection index response (USD 10/feed unit) were found for direct selection (***I*_2_**) compared with indirect selection (***I*_8_**). For RFI, higher genetic and economic gains for MY, FY, and DO; lower gains for PY and PL; higher index accuracy (1 percentage point); and equal total selection index response were found for direct selection (***I*_3_**) compared to indirect selection (***I*_9_**). Regardless of direct or indirect selection for feed efficiency, the total selection index response was generally higher if RFI rather than DMI was used in the selection index.

### 3.4. Feed Efficiency Sub-Index

The genetic regression coefficients for MY, FY, PY, and FSAV included in FE$ were estimated to be 0.015, 0.451, 0.518, and 1.000, respectively. Multiplying those coefficients by the EVs of the traits resulted in EW estimates of 0.002 for MY, 1.42 for FY, 1.27 for PY, and 0.762 for FSAV, as well as a conversion Equation (1) of ${\mathrm{PTA}}_{{\mathrm{FE}\$}_{\mathrm{Iran}}}=-44.24+1.262\left({\mathrm{PTA}}_{{\mathrm{FE}\$}_{\mathrm{USA}}}\right)$.

## 4. Discussion

### 4.1. Meta-Analysis

The meta-analysis in this study evaluated the extent of utilizable genetic variation in feed intake, production, reproduction, type, BW, and BCS traits of Holstein dairy cows. The heritability estimates reported in the studies were 0.10 for DMI [35], from 0.15 to 0.18 for 305-day feed intake traits [36], from 0.21 to 0.40 for DMI across lactations [10], 0.27 for lactation total DMI [37], and from 0.09 to 0.38 for RFI [14,38]. Those heritability estimates indicated that the regulation of feed intake has a substantial genetic component and can be successfully altered through selection [39]. Difficulty in accurately estimating genetic parameters for feed intake traits could result from the limited number of observations recorded per year or the lack of a suitable statistical model. As a reproductive trait, DO was generally reported with low heritability and strongly influenced by herd management, climate differences, and level of milk production [40,41,42,43]. As a longevity trait, PL is defined as the length of time from the first day of calving to culling or death of the cow. Although PL is primarily affected by environmental effects such as year and age at first calving, season, and herd, it also has a genetic component. Most studies reported low heritability for PL, but estimates depended on the model used for estimating genetic variation [44,45,46]. Due to the fact that heritability estimates for type traits generally ranged from moderate to high, indirect selection could be useful for economic traits of interest that have a strong genetic correlation with type traits [14,18,47,48,49]. Heritability estimates for BCS ranged from 0.17 to 0.34 [11,14,48,49]. Several studies reported that the highest heritability for BCS occurred when mean BCS was lowest; BCS tends to be lowest in early lactation when it is less influenced by differences in management practices [50,51,52]. For BW, heritability estimates ranged from 0.35 to 0.71 [11,14,36,38] and varied at different stages of a dairy cow’s life [53].

Meta-analysis revealed a weak genetic correlation of RFI with production traits, which is due to the fact that RFI is genetically independent of production traits, [13,14,18,54], indicating that the indirect selection for production traits will not improve RFI but will enhance DMI due to a strong genetic correlation of DMI with production traits [9,14,36]. However, the strong positive genetic correlation between RFI and DMI [14,38] suggests that selection of low RFI can improve the feed efficiency of dairy cattle. For reproductive traits, the low negative genetic correlation between DMI and DO (−0.14 ± 0.29) reported by [55] indicated that higher-yield cows had higher DMI and longer DO or lower reproductive performance. The same study reported a high negative genetic correlation between DO and RFI (−0.51 ± 0.76), indicating that cows with low RFI have longer DO and consequently poorer reproductive performance. However, standard errors for this genetic correlation were high, and additional research with a larger dataset is needed to confirm this interpretation. In contrast, Coleman et al. [56] reported no association between fertility traits and RFI. The genetic correlation between feed efficiency traits (DMI and RFI) and PL indicates that the selection of DMI against RFI can lead to longer stays of dairy cows in the herd. Although low heritability, voluntary culling, and post-deletion records can affect genetic gain for PL, this longevity trait has received much more attention in the past decades due to its high economic value and direct impact on profitability [57]. In addition, selecting dairy cattle for longevity traits in herds can reduce the cost of heifer replacements and increase the PL of high-yielding cows [58].

In most dairy cattle breeding programs, type traits are recorded because they are easy and affordable to measure and are heritable [59,60]. Several studies reported medium-to-high heritability for type traits [14,37,49,61], which agreed with the meta-analysis heritability estimate. Some of the type traits in this study, such as chest width, angularity, and stature, were used as indicator (or indirect) traits to select DMI and RFI in dairy cows due to the medium-to-high genetic correlations between those traits. The magnitude and direction of genetic correlation between body size traits (BCS and BW) and feed efficiency traits (DMI and RFI) showed that the genetic variation of DMI and RFI in dairy cows is explained better by BW than by BCS. Therefore, BW can be used as an indirect trait to select for DMI and RFI.

### 4.2. Economic Weights

Despite the same trait definition and pricing system for milk yield, the estimated EVs for production traits in this study differed from previous studies [28,62]. Sadeghi-Sefidmazgi et al. [28] reported EVs of USD 0.15/kg/cow/year for MY, USD 1.36/kg/cow/year for FY, and USD −1.02/kg/cow/year for PY in contrast to USD 0.34, USD 6.93, and USD 5.53, respectively (Table 5); Ghiasi et al. [62] reported an EW for MY of USD 0.193/kg/cow/year. The variation in the EVs of MY is the result of differences in the sale price of milk and production costs, including feed and labor costs. The EV of DO for all milk producing abilities was reported to be positive (from USD 0.21 to USD 0.40) when calving interval was increased from 12 to 13 month and negative (USD −0.04 to USD −0.23) when it was increased from 13 to 15 month [63]. Recently, the EV of DO was evaluated for Japanese Holstein cows and ranged from USD 3.59 to 7.72 [64], which is not similar to the estimated EV for DO of USD −2.71 in this study (Table 5). Sadeghi-Sefidmazgi et al. [28] reported an EV of USD 10.13/month for PL, which is higher than the EV reported in this study (USD 2.77/month/cow/year) (Table 5). Differences in maintenance costs as well as profits obtained through PL vary among different economic conditions around the world, which is the primary reason for differentiating EV from EW among studies.

To assess the economic importance of feed efficiency in the breeding objective for dairy cattle, the EWs of feed efficiency traits should be determined under current Iranian dairy production conditions. In this study, the different EWs for DMI estimated in different scenarios showed that the EW was highly sensitive to feed and animal factors influencing the feed intake of dairy cows as well as the difference between concentrate and roughage price. This sensitivity requires a consideration of production circumstances in defining breeding objectives. 

The calculation of EVs for RFI was based on a daily DMI increase or decrease of 1 kg/day/animal. Therefore, the EV for RFI shows how the profit of a production system expressed per cow per year would decrease if the daily DMI of cows increased by 1 kg/day/cow over the whole rearing period, and the unit for RFI is monetary units/cow/year. Hietala et al. [65] calculated EWs for additional feed efficiency traits of Finnish Ayrshires, and the marginal EVs for RFI of breeding heifers and cows were estimated to be USD −30.6 and USD −66.96/kg DMI/day/cow/year, respectively. Krupová et al. [3] estimated the marginal EVs for RFI to be USD −66.18 and USD −65.568/kg DMI/day/cow/year for cows and breeding heifers in the dairy system, respectively, and USD −24.54, USD −13.56, and USD −7.248/kg DMI/day/cow/year for cows, breeding heifers, and fattened animals in the cow-calf system, respectively. In the 2021 US net merit index [6], the EV for RFI was estimated to be USD −0.14/kg of PTA RFI. The estimated EV for feed intake in this study was in the range of the estimated EV of other studies. The variation in the EVs for RFI can be attributed to differences in trait definition and expression of unit traits as well as assumptions related to management systems.

### 4.3. Selection Index Scenarios

In most countries, production traits were included as the initial economic traits in selection indices for dairy cattle. During the last 2 decades, breeding objectives have been extended to longevity, reproduction, health, and functionality traits for dairy cattle breeding programs worldwide [60,66,67]. The importance of integrating feed efficiency traits in Iranian breeding goals for genetic improvement has been determined due to the fact that studies in several countries have shown the substantial impact on the profitability of dairy production. Although data on feed intake in the dairy production system of Iran are missing and not recorded, forming a selection index using a meta-analysis may be valuable. Consequently, the importance of including two feed efficiency traits (DMI and RFI) in breeding objectives was investigated through meta-analysis in this study.

Based on the selection criteria defined for the first scenario, the direct selection for DMI (***I*_2_**) resulted in higher economic and genetic gains for MY and FY than did the direct selection for RFI (***I*_3_**), whereas the direct selection for RFI resulted in higher economic and genetic gains for PY, PL, and DO (Table 6). Due to the fact that RFI is genetically independent of production, its use as a selection criterion is equivalent to using restricted selection criteria to stabilize the production constant [21]. An unfavorable genetic and economic trend was observed for DO if either DMI or RFI were included in the selection index. The presence of antagonistic genetic correlations among traits included in an index is crucial for the response to selection. Additionally, the genetic improvement for DO relies on the heritability and economic weight of all traits incorporated in the index. Therefore, anticipating genetic gains based on a single affecting factor is not sufficient. Cows with low RFI can easily mobilize body reserves if changes in BCS are not included in the prediction of RFI or if BCS does not show all changes in body tissue mobilization. Due to a modest unfavorable genetic correlation between RFI and fertility, mobilization of body reserves has a negative effect on fertility [68,69,70,71]. The results of this study are consistent with the results of previous studies that reported unfavorable genetic correlations between RFI and fertility measures [55] and between DMI and DO [72]. Profit per feed unit (total selection index response) was higher when BW or BCS was substituted for DO in selection indices ***I*_4_** through ***I*_7_** than for the baseline index ***I*_1_** (Table 6). Selection criteria of MY, FY, PY, PL, BW, and RFI were determined to result in the most profitable index (***I*_6_**) when compared to others in the first scenario and used a suitable profit function to estimate the relative EW of the traits. Including the direct selection for RFI in ***I*_6_** resulted in the highest total selection index response, which has also been reported by [73]. 

The direct selection for feed intake is not commonly used around the world because collecting feed intake data in most commercial herds is costly [74]. In several studies, phenotypic [75,76] and genetic [36,47,77,78] correlations between feed intake and type traits have been reported. An alternative to direct selection on feed intake data is to use indirect traits such as type traits to select for feed efficiency in dairy cows. Indirect selection on DMI or RFI was examined using ***I*_8_** and ***I*_9_**, respectively. Genetic and economic gains for traits in the breeding objectives were generally higher for ***I*_8_** than for ***I*_9_** (Table 6). However, the total selection index response for ***I*_8_** was USD 20/feed unit less than that for ***I*_9_**, and neither index had a total response higher than that for the baseline index.

The inclusion of RFI or DMI in breeding objectives in the absence of a robust genetic correlation between other breeding objective traits results in unfavorable selection responses. In a study of Australian dairy systems across breeds, increasing survival and reducing milk volume (i.e., increasing fat and protein content), live weight, DMI, SCC, and poor fertility were the only combination of traits that could increase net income and reduce greenhouse gas emissions per cow and per unit of milk [79]. In addition, investigation of the economic effect of genomic selection for RFI indicated that integrating genomic breeding values for RFI in the Australian Profit Ranking selection index would improve the annual rate of gain in profitability for Australian dairy cattle by 3.8% [80] FSAV has been included in two Australian selection indices (the Balance Performance Index, BPI; and the Health Weighted Index, HWI) since 2015 and Bolormaa et al. [81] reported that the reliability of lifetime RFI using trivariate analysis increased from 11% to 20% compared with the 2015 model, indicating that the genetic trend for FSAV is expected to shift in a favorable direction when FSAV is included in those indices. In our study, both the direct and indirect selection on RFI in a genetic–economic index were economically more profitable than direct or indirect selection on DMI. The use of RFI has been preferred because it accounts for milk production, body size traits, and parity [82] and is a potential target for genetic selection due to its connection to herd profitability and decreased environmental impact of dairy cattle. A study by Marinho et al. [83] showed that the most feed-efficient cows ate approximately 4 kg/day less dry matter and produced the same amount of energy-corrected milk (ECM) with improved reproductive performance compared with the least feed-efficient cows. 

### 4.4. Feed Efficiency Sub-Index

In April 2021, the EWs for MY, FY, PY, and FSAV (the four main components in FE$) were estimated to be USD 0.0018/kg, USD 3.42/kg, USD 3.82/kg, and USD 0.24/kg, respectively [34]. The discrepancy between the Iranian and US EW for each trait in FE$ can be attributed to differences in genetic and phenotypic parameters and EV for each trait. To account for differences in Iranian and US dairy production and economic conditions, the PTA FE$ of US bulls should be multiplied by the conversion coefficients found for (1) to estimate Iranian PTA FE$. A more precise Iranian EV for each trait in FE$ can be determined by collecting DMI data from Iranian herds and initiating genetic evaluation for RFI.

## 5. Conclusions

Including feed efficiency in breeding objectives is desirable due to the potential benefit for increased sustainability and herd profitability. An investigation of different selection indices that included direct and indirect selection for feed efficiency traits showed that the best total selection index response was observed when RFI was directly included as a selection criterion. Although the inclusion of RFI or DMI in selection criteria is recommended, using indicator traits (e.g., type traits) could be a useful proxy to select for feed efficiency in the absence of direct genetic evaluation for feed efficiency traits. Breeding objectives of the current Iranian selection index, which currently focus on future production and marketing environment, should include feed efficiency in future evaluations.

## Figures and Tables

**Table 1 animals-13-01216-t001:** Weighted mean heritabilities and summary statistics for feed efficiency, production, reproduction, longevity, type, body weight, and body condition score traits calculated by meta-analysis across dairy cattle breeds and countries.

Trait	N ^1^	Weighted Mean h^2^	SE	Range of h^2^ in the Literature
DMI, kg/day	10 (11)	0.21	0.07	0.08 to 0.52
Residual feed intake, kg/day	5 (7)	0.19	0.09	0.09 to 0.38
Milk, kg	21 (24)	0.22	0.03	0.12 to 0.39
Fat, kg	15 (16)	0.22	0.05	0.06 to 0.25
Protein, kg	15 (16)	0.22	0.04	0.13 to 0.36
Productive life, month	2 (2)	0.10	0.03	0.01 to 0.11
Days open, day	5 (4)	0.03	0.01	0.002 to 0.03
Stature, cm/scale	5 (7)	0.46	0.01	0.40 to 0.60
Chest width, scale	3 (5)	0.26	0.01	0.19 to 0.31
Body depth, scale	5 (7)	0.39	0.01	0.28 to 0.41
Angularity, scale	4 (6)	0.23	0.01	0.21 to 0.29
Rump angle, scale	3 (5)	0.37	0.06	0.20 to 0.42
Rump width, scale	2 (4)	0.27	0.05	0.18 to 0.40
Body weight, kg	5 (7)	0.58	0.04	0.35 to 0.71
Body condition, score	4 (5)	0.23	0.02	0.17 to 0.34

^1^ Number of articles with number of databases in parentheses.

**Table 2 animals-13-01216-t002:** Weighted mean genetic correlations and summary statistics between feed efficiency traits and production, reproduction, longevity, type, body weight, and body condition score traits calculated by meta-analysis for Holsteins across countries.

Feed Efficiency Trait	Correlated Trait	N ^1^	Weighted Mean Genetic Correlation	SE	Range of Genetic Correlations in the Literature
Dry matter intake	Residual feed intake	3 (5)	0.53	0.07	0.38 to 0.89
	Milk yield	8 (9)	0.68	0.08	0.36 to 0.78
	Fat yield	4 (4)	0.51	0.07	0.15 to 0.53
	Protein yield	4 (4)	0.55	0.08	0.25 to 0.56
	Productive life	1 (2)	0.49	0.08	0.48 to 0.51
	Days open	1 (2)	−0.14	0.09	−0.14 to −0.15
	Stature	5 (6)	0.44	0.01	0.32 to 0.57
	Chest width	2 (3)	0.55	0.09	0.45 to 0.68
	Body depth	4 (5)	0.40	0.09	0.26 to 0.49
	Angularity	3 (4)	0.58	0.09	−0.02 to 0.60
	Rump angle	1 (2)	0.15	0.04	0.10 to 0.21
	Rump width	2 (3)	0.12	0.07	0.04 to 0.18
	Body weight	4 (6)	0.52	0.01	0.35 to 0.71
	Body condition score	4 (5)	0.36	0.08	−0.04 to 0.71
Residual feed intake	Milk yield	4 (5)	0.08	0.05	−0.05 to 0.35
	Fat yield	3 (3)	−0.01	0.07	−0.07 to 0.20
	Protein yield	3 (3)	−0.04	0.09	−0.03 to −0.06
	Productive life	1 (2)	−0.23	0.09	−0.22 to −0.24
	Days open	1 (2)	−0.50	0.04	−0.48 to −0.51
	Stature	2 (3)	0.28	0.09	0.12 to 0.43
	Chest width	2 (3)	0.17	0.09	0.06 to 0.39
	Body depth	2 (3)	0.08	0.09	0.05 to 0.11
	Angularity	2 (3)	0.14	0.09	0.08 to 0.41
	Rump angle	1 (2)	−0.06	0.04	−0.06 to −0.07
	Rump width	1 (2)	−0.03	0.01	−0.02 to −0.06
	Body weight	2 (3)	0.15	0.07	0.03 to −0.26
	Body condition score	2 (3)	−0.13	0.04	−0.14 to 0.46

^1^ Number of articles with number of databases in parentheses.

**Table 3 animals-13-01216-t003:** Heritabilities (diagonal, bold) and phenotypic (above diagonal), and genetic (below diagonal) correlations of feed efficiency, production, reproduction, longevity, type, body weight, and body condition score traits used from meta-analysis to calculate optimal selection index weights for selection criteria formulas.

Trait ^1^	DMI	RFI	MY	FY	PY	PL	DO	ST	CW	ANG	BW	BCS
DMI	**0.21**	0.75	0.66	0.60	0.67	0.03	0.09	0.22	0.35	0.24	0.35	−0.02
RFI	0.53	**0.19**	−0.07	−0.05	−0.09	−0.06	−0.02	0.12	0.17	0.12	0.01	0.04
MY	0.68	0.08	**0.22**	0.69	0.85	0.15	0.29	0.10	0.13	0.10	0.09	−0.23
FY	0.51	−0.01	0.41	**0.22**	0.60	0.23	0.14	0.01	0.08	0.19	0.14	−0.15
PY	0.55	−0.04	0.96	0.56	**0.22**	0.26	0.13	0.13	0.02	0.21	0.20	−0.16
PL	0.49	−0.23	0.64	0.54	0.63	**0.10**	−0.10	−0.03	−0.03	0.01	−0.02	0.00
DO	−0.14	−0.50	0.42	0.32	0.35	−0.60	**0.03**	0.08	−0.03	0.01	−0.07	−0.17
ST	0.44	0.28	0.12	0.10	0.12	−0.19	0.17	**0.46**	−0.01	0.06	0.38	0.08
CW	0.55	0.17	0.12	0.10	0.12	−0.19	0.17	0.05	**0.26**	−0.11	0.45	0.24
ANG	0.58	0.14	0.26	0.13	0.14	−0.13	0.42	0.10	−0.03	**0.23**	−0.07	−0.20
BW	0.52	0.15	−0.29	−0.03	0.03	−0.22	−0.23	0.94	0.84	−0.18	**0.58**	0.31
BCS	0.36	−0.13	−0.30	−0.27	−0.31	−0.48	−0.23	−0.13	0.72	−0.65	0.85	**0.23**

^1^ DMI= dry matter intake, RFI = residual feed intake, MY = milk yield, FY = fat yield, PY = protein yield, PL = productive life, DO = days open, ST = stature, CW = chest width, ANG = angularity, BW = body weight, BCS = body condition score.

**Table 4 animals-13-01216-t004:** Mean values of production data, prices, and costs of each unit of the variables considered for computing economic values.

Variable	Mean
**Production data**	
305-day milk, kg	11,750
305-day fat, kg	399.5
305-day protein, kg	364.3
Longevity, year	3.86
Age at first calving, month	24
Calving interval, day	420
Stillbirth rate, %	0.04
Calf mortality, %	0.01
Productive cow mortality, %	0.01
Heat detection rate	0.50
Estrous cycle, day	21
Conception rate	0.47
Pregnancy rate	0.23
Live weight of culled calf, kg	240
Live weight of culled heifer, kg	500
Live weight of culled cow, kg	680
Proportion of male calves sold to a feedlot	0.41
Male and female calves reared until 3 month of age/cow/year, n	0.83
Female calves reared/cow/year, n	0.40
Involuntary culling rate of calves from birth until 3 month of age	0.005
Survival rate of heifers from 3 month of age until calving	0.98
Involuntary culling rate of heifers as a proportion of surplus heifers	0.003
**Prices**	
Base milk, USD/kg	1.07
Accessory payment for milk fat, USD/kg	0.14
Accessory payment for milk protein, USD/kg	0.14
Male calf price, USD/calf	1071
Female calf price, USD/calf	1429
Replacement heifer, USD/heifer	6667
Culled calves, USD/kg	7.14
Culled heifers, USD/kg	7.14
Culled cows, USD/kg	5.95
Conventional domestic semen dose, USD	12.14
Conventional imported semen dose, USD	24.05
Sexed imported semen dose, USD	80.95
Insemination, USD	26.89
**Costs**	
Base milk, USD/kg	0.75
Milk fat accessory, USD/kg	7.36
Milk protein accessory, USD/kg	7.37
NEL, USD/Mcal	0.79
MP, USD/kg	0.002
Calf rearing, USD/calf	1000
Rearing from 3 month of age until calving, USD/heifer	4000
Rearing from 3 to 21 month of age, USD/heifer	3600

**Table 5 animals-13-01216-t005:** Economic values, genetic expressions, and estimated economic weights for each breeding objective trait.

Trait	Economic Value (USD/Unit of Trait/Cow/year)	Genetic Expression	Economic Weight (USD/Unit of Trait)
Milk, kg	0.34	1.00	0.34
Fat, kg	6.93	1.00	6.93
Protein, kg	5.53	1.00	5.53
Days open, day	−2.71	1.00	−2.71
Productive life, month	2.77	0.17	0.47
DMI, kg	−1.68	1.00	−1.68
Residual feed intake, kg	−1.70	1.00	−1.70

**Table 6 animals-13-01216-t006:** Expected genetic and economic gains for breeding objective traits based on 9 selection indices, index accuracy, and total index response.

Trait	Selection Index ^1^								
	*I* _1_	*I* _2_	*I* _3_	*I* _4_	*I* _5_	*I* _6_	*I* _7_	*I* _8_	*I* _9_
Genetic gain (USD/unit gain)									
Milk yield, kg	658.88	668.30	664.40	649.61	655.34	708.77	657.67	659.64	656.76
Fat yield, kg	12.35	13.03	12.84	11.80	12.45	13.57	12.47	12.45	12.53
Protein yield, kg	16.94	16.16	16.66	17.19	17.01	15.78	16.97	16.92	16.86
Productive life, month	3.07	2.58	2.61	3.13	3.06	2.90	3.13	2.91	2.90
Days open, day	2.27	−0.43	0.28	−1.80	0.72	1.80	0.31	2.12	1.90
DMI, kg/day	−	10.30	−	−2.70	−7.01	−	−	9.77	−
Residual feed intake, kg/day	−	−	−0.77	−	−	−5.26	−7.03	−	−2.10
Economic gain (USD/unit gain)									
Milk yield, kg	224.00	227.20	225.90	220.90	222.08	241.00	223.60	224.30	223.30
Fat yield, kg	85.64	90.30	89.00	81.78	86.30	94.06	86.47	86.28	86.89
Protein yield, kg	93.70	89.34	92.15	95.08	94.07	87.30	93.87	93.60	93.28
Productive life, month	8.50	7.13	7.28	8.68	8.49	8.03	8.67	8.07	8.05
Days open, day	−6.14	1.16	−0.78	4.88	−1.96	−4.88	−0.85	−5.75	−5.17
DMI, kg/day	−	−17.31	−	4.53	11.78	−	−	−16.42	−
Residual feed intake, kg/day	−	−	1.31	−	−	8.95	11.97	−	3.58
Index accuracy	0.89	0.92	0.91	0.96	0.97	0.96	0.93	0.90	0.90
Total selection index response, (USD/unit gain)	410	400	410	420	420	430	420	390	410

^1^ Traits included in indices: ***I*_1_** (milk yield, fat yield, protein yield, productive life, and days open), ***I*_2_** (milk yield, fat yield, protein yield, productive life, days open, and DMI), ***I*_3_** (milk yield, fat yield, protein yield, productive life, days open, and residual feed intake), ***I*_4_** (milk yield, fat yield, protein yield, productive life, body weight, and DMI), ***I*_5_** (milk yield, fat yield, protein yield, productive life, body condition score, and DMI), ***I*_6_** (milk yield, fat yield, protein yield, productive life, body weight, and residual feed intake), ***I*_7_** (milk yield, fat yield, protein yield, productive life, body condition score, and residual feed intake), ***I*_8_** (milk yield, fat yield, protein yield, productive life, days open, chest width, and angularity), and ***I*_9_** (milk yield, fat yield, protein yield, productive life, days open, chest width, and stature).

## Data Availability

The data supporting the findings of this study are available upon request from the author Ali Sadeghi-Sefidmazgi: sadeghism@ut.ac.ir and with permission from the Animal Breeding Center of Iran (ABCI).

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
