# Peer review of "Implementation of Feed Efficiency in Iranian Holstein Breeding Program"

_animals, 2023, doi:10.3390/ani13071216_

Round 1

Reviewer 1 Report

This is an interesting and well presented study, but a lack of context makes it impossible to interpret how the results might apply elsewhere.

My main concern is that I don't know anything about the 7 large Holstein dairy herds other than they contain > 1000 milking cows.  This makes it difficult for me to understand how (and whether) the data presented here relates to Iranian forms generally, or any particular farm type.  The reference given to explain more about the farms [28] is from 2012.  The authors could do well to use this reference as a template for the type of background information that is needed for an international audience to understand the context of the work.  How were the farms selected?  Are they typical of Iranian farms? What was the range of cattle, breeds, and genetic merit of the farms chosen.  I would expect the results might have been different  if very high or low genetic merit farms were used.  

I'm also interested that genomics isn't more of a focus in Iran as it is in other countries - a little more explanation about wqhy this is the case would be nice.

Author Response

We would like to thank you and the reviewers for your helpful suggestions and the opportunity to improve our manuscript. All questions and suggestions made by the reviewers were addressed. The reviewers’ comments are listed in the order they were received. First, the reviewer question (Q#:) is listed in italic font followed by our response in regular font (AU#). For minor or specific comments/suggestions, the reviewer comment is listed first in italic font and underlined followed by our response in regular font. Changes made to the manuscript are using the “Track changes” function.

Response to Reviewer #1

Q1: My main concern is that I don't know anything about the 7 large Holstein dairy herds other than they contain > 1000 milking cows.  This makes it difficult for me to understand how (and whether) the data presented here relates to Iranian forms generally, or any particular farm type.  The reference given to explain more about the farms [28] is from 2012.  The authors could do well to use this reference as a template for the type of background information that is needed for an international audience to understand the context of the work.  How were the farms selected?  Are they typical of Iranian farms? What was the range of cattle, breeds, and genetic merit of the farms chosen.  I would expect the results might have been different if very high or low genetic merit farms were used.

AU: Your thoughts on our work are greatly appreciated and your insightful suggestions were incorporated as best we could. The changes made to reflect the reviewer’s comment (L143-150 in the revised manuscript).  

Q2: I'm also interested that genomics isn't more of a focus in Iran as it is in other countries - a little more explanation about why this is the case would be nice.

AU: Genomics is currently considered an expensive technology in Iran, but it is expected to become more cost-effective in the future. Presently, approximately one-third of AI bull semen utilized in the country is imported from western countries. However, with an improvement in knowledge and a great gene pool, there is a possibility for mutual gene flow with exporting nations.

Reviewer 2 Report

Please find below my minor suggestions:

L59: 

"An alternative definition of RFI is that RFI has been genetically adjusted for the main energy sinks (e.g., milk yield, BW), making it genetically uncorrelated with those energy sink traits."

As RFI has been genetically adjusted for the main energy sinks (e.g., milk yield, BW), it is genetically uncorrelated with those energy sink traits.

L65:

"The heritability of RFI is depends"

The heritability of RFI depends

L65:

"The heritability of RFI is depends largely on the environmental correlation between feed intake and milk production."

Could you please elaborate on this argument? What was the point here? 

L66:

"RFI based on phenotypic and genotypic regression of feed intake on milk production is dependent and independent of production, respectively [21]."

This sentence is a bit confusing. RFI based on regression using phenotypes would be phenotypically independent of the model factors. Please consider rewriting this sentence or removing it. 

L150:

"cumulative discounted expressions"

Please clarify. 

L163:

Add "SE = " in the equation 

L141-143:

Please review this sentence. 

L147:

"7 large Holstein dairy"

7 Holstein dairy

L178:

"Economic traits ..."

Do you mean economic index? Please review this sentence.

L178:

What do you mean by "aggregate genotypes". Please clarify 

L248:

Add "b = " in the equation

L314:

I'd recommend a full stop (.) instead of ";" here. 

L345:

"is higher"

was higher 

Discussion:

This paper might be helpful when discussing genetic parameters for economically important traits (production, type, fertility). 

https://www.sciencedirect.com/science/article/pii/S0022030221005749

L386:

"RFI genetically is being independent of production, [13,14,18,53], ...."

RFI is genetically independent of production traits [13,14,18,53], .....

L390:

"low RFI might improve the"

low RFI can improve the

L405-414:

It's not clear to me how these type traits could be used as a proxy to select for Feed Efficiency. I understood that they have a favorable genetic correlation. On L413, when you said "BW can be used as an indirect trait to select for DMI and RFI", are suggesting to select for weight gain, seeking animals with lower gain? I'm wondering what would be the impact of this on other traits (e.g. helth, fertility, production, etc). Also, what would be the best moment to measure the body weight of an animal?

L470:

"A reduction in genetic and"

Please use 'favorable' or 'unfavorable'. In this case, it's not clear if this reduction was desirable or not.  

L476:

"reported negative genetic correlations between"

Please use 'favorable' or 'unfavorable'.

L493" 

Shoud it be "RFI and DMI" or "RFI or DMI" ? 

L501

"by 3.8% [79] (Hayes et al., 2011)"

by 3.8% [79]

Author Response

We would like to thank you and the reviewers for your helpful suggestions and the opportunity to improve our manuscript. All questions and suggestions made by the reviewers were addressed. The reviewers’ comments are listed in the order they were received. First, the reviewer question (Q#:) is listed in italic font followed by our response in regular font (AU#). For minor or specific comments/suggestions, the reviewer comment is listed first in italic font and underlined followed by our response in regular font. Changes made to the manuscript are using the “Track changes” function.

Response to Reviewer #2

Q1: "An alternative definition of RFI is that RFI has been genetically adjusted for the main energy sinks (e.g., milk yield, BW), making it genetically uncorrelated with those energy sink traits." As RFI has been genetically adjusted for the main energy sinks (e.g., milk yield, BW), it is genetically uncorrelated with those energy sink traits.

AU: Change made (lines 61-63 in revised manuscript).

Q2: L65 "The heritability of RFI is depends", The heritability of RFI depends.

AU: Change made (line 67 in revised manuscript).

Q3: L65: "The heritability of RFI is depends largely on the environmental correlation between feed intake and milk production." Could you please elaborate on this argument? What was the point here?

AU: The heritability of RFI is subject to significant variation depending on the genetic and phenotypic parameters of the related traits. In cases where phenotypic regressions are employed, the heritability of RFI is largely dependent on the environmental correlations between feed intake and other associated traits, which tend to be more influential than genetic correlations. Although RFI is independent of other related traits, except for feed intake, it is not genetically independent, and the magnitude and sign of the genetic correlations are influenced by both genetic and environmental correlations with feed intake. In general, low genetic correlation and high environmental correlation between feed intake and related traits are favorable.

Q4: L66: "RFI based on phenotypic and genotypic regression of feed intake on milk production is dependent and independent of production, respectively [21]."This sentence is a bit confusing. RFI based on regression using phenotypes would be phenotypically independent of the model factors. Please consider rewriting this sentence or removing it.

AU: Change made (lines 68-70 in the revised manuscript).

Q5: L150: "Cumulative discounted expressions". Please clarify.

AU: Clarification made. Please see L163-166 in the revised manuscript.

Q6: L163: Add "SE = " in the equation.

AU: Change made (line 128 in the revised manuscript).

Q7: L141-143: Please review this sentence.

AU: Change made (lines 143-150 in the revised manuscript).

Q7: L147: "7 large Holstein dairy", 7 Holstein dairy.

AU: Change made (lines 146 in the revised manuscript).

Q8: L178: "Economic traits… ". Do you mean economic index? Please review this sentence.

AU: Change made for clarification (lines 193-196 in the revised manuscript).

 Q9: L178: What do you mean by "aggregate genotypes". Please clarify.

AU: aggregate genotypes, breeding objective or economic selection index are the same and is followed by the formula where  is economic weight for i trait and  is estimated breeding value for i trait. For consistency, aggregate genotypes were replaced with breeding objective.

Q10: L248: Add "b = " in the equation.

AU: Change made (line 265 in the revised manuscript).

Q11: L314: I'd recommend a full stop (.) instead of ";" here.

AU: Change made (line 331 in the revised manuscript).

Q12:L345: "is higher", was higher

AU: Change made (line 364 in the revised manuscript).

Q13: Discussion: This paper might be helpful when discussing genetic parameters for economically important traits (production, type, fertility). https://www.sciencedirect.com/science/article/pii/S0022030221005749.

AU: Thanks for your suggested paper. We have used some information from this paper.

Q14:L386: "RFI genetically is being independent of production, [13,14,18,53], ...."

RFI is genetically independent of production traits [13,14,18,53], .....

AU: Change made (line 406 in the revised manuscript).

Q15:L390: "low RFI might improve the", low RFI can improve the

AU: Change made (line 410 in the revised manuscript).

Q15: L405-414: It's not clear to me how these type traits could be used as a proxy to select for Feed Efficiency. I understood that they have a favorable genetic correlation. On L413, when you said "BW can be used as an indirect trait to select for DMI and RFI", are suggesting to select for weight gain, seeking animals with lower gain? I'm wondering what would be the impact of this on other traits (e.g. helth, fertility, production, etc). Also, what would be the best moment to measure the body weight of an animal?

AU: BW as an estimate of maintenance requirements and RFI were used to identify metabolically efficient cows in the indices. Thus, selecting for smaller animals, with negative body weight composite, is expected to reduce maintenance requirements that should improve feed efficiency. Considering the genetic correlation between feed efficiency traits with production traits, health and fertility, these traits should be carefully controlled to ensure that the gain in feed efficiency is not accompanied by the loss of health, fertility and productive life. Cows with negative RFI can easily mobilize body reserves if changes in BCS are not included in the prediction of RFI, or if BCS does not show all changes in body tissue mobilization. Due to the modest unfavorable genetic correlation between RFI and fertility, mobilization of body reserves has a negative impact on fertility. Despite the negative correlation between RFI and DO, the researchers wanted to investigate the effect of removing DO trait on increasing productive and feed efficiency traits.

Considering the complexity of the metabolic changes of cows across lactation stages, it is possible that the partial regression coefficients of feed intake on milk production, body maintenance, and ΔBW could vary across lactation stages. Therefore, the general RFI model with constant partial regression coefficients of feed intake on energy sinks might not always hold for the entire lactation period, which in turn might influence the estimation of RFI. An alternative definition of RFI is to derive RFI that is genetically adjusted for the major energy sinks (e.g., milk yield, BW), so that RFI is genetically uncorrelated with major energy sink traits. In dairy cattle breeding, it is of interest to define RFI that is genetically uncorrelated with major energy sink traits (e.g., milk yield, BW). Milk production traits and BW-related traits are often part of the total merit index traits in the genetic evaluation of dairy cattle. After eliminating the genetic correlations of RFI with milk production and BW, RFI becomes a more independent trait for representing feed efficiency in the selection index.

Q16: L470: "A reduction in genetic and". Please use 'favorable' or 'unfavorable'. In this case, it's not clear if this reduction was desirable or not. 

AU: Change made (lines 489-494 in the revised manuscript).

Q17: L476: "Reported negative genetic correlations between". Please use 'favorable' or 'unfavorable'.

AU: Change made (line 499 in the revised manuscript).

Q18: L493" Should it be "RFI and DMI" or "RFI or DMI"?

AU: Change made for more clarification (line 516 in the revised manuscript).

Q19:L501: "by 3.8% [79] (Hayes et al., 2011)", by 3.8% [79]

AU: Change made (line 525 in the revised manuscript).

Reviewer 3 Report

Dear Authors,

this a well-written and interesting paper. The aim is certainly of interest as it is an hot-topic for the livestock sector.

I have just a couple of minor comments and suggestions.

Lines 50-68: some authors focused only on the concentrate intake, since it is the most expensive part of the animals' rations and because sometimes we don't have information about the amount of hay (10.1080/1828051X.2021.1963864)

Line 305, Table 3: you should explain why there are no values for one combination, chest width vs BCS. 

Lines 314-315: it was hard to understand what genetic expression means.

Author Response

We would like to thank you and the reviewers for your helpful suggestions and the opportunity to improve our manuscript. All questions and suggestions made by the reviewers were addressed. The reviewers’ comments are listed in the order they were received. First, the reviewer question (Q#:) is listed in italic font followed by our response in regular font (AU#). For minor or specific comments/suggestions, the reviewer comment is listed first in italic font and underlined followed by our response in regular font. Changes made to the manuscript are using the “Track changes” function.

Response to Reviewer #3

Q1: Lines 50-68: some authors focused only on the concentrate intake, since it is the most expensive part of the animals' rations and because sometimes we don't have information about the amount of hay (10.1080/1828051X.2021.1963864).

AU: That is correct. In Iran, the prices of forage and concentrate feedstuff are strongly correlated. Although we import soybeans, corn, and barley grains, we cultivate alfalfa and corn silage. Therefore, when making economic calculations, it is crucial to consider both components separately.

Q2: Line 305, Table 3: you should explain why there are no values for one combination, chest width vs BCS.

AU: The association between these two traits was not explored in the meta-analysis studies. While conducting the simulation study, we did not require the values between these two traits. However, to provide a comprehensive table, we obtained the values from the https://www.sciencedirect.com/science/article/pii/S0022030221005749 and incorporated them into Table 2.

Q3: Lines 314-315: it was hard to understand what genetic expression means.

AU: Clarification made. Please see L163-166 in the revised manuscript.